# How much we express love predicts how much we feel loved in daily life

Lindy Williams[1], Sharon H. Kim[1], Yanling Li[1], Saida Heshmati[2], Joachim Vandekerckhove[3], Robert W. Roeser[1], Zita Oravecz [1,4]*

1 Human Development and Family Studies, The Pennsylvania State University, State College, Pennsylvania, United States of America, 2 Department of Psychology, Claremont Graduate University, Claremont, California, United States of America, 3 Cognitive Sciences, University of California, Irvine, California, United States of America, 4 Institute for Computational and Data Sciences, The Pennsylvania State University, State College, Pennsylvania, United States of America

* zita@psu.edu

## Abstract

Feeling and expressing love in daily life are interconnected and perhaps mutually influential experiences. In this study we examined the reciprocal dynamics of feeling and expressing love and its relation to well-being using an ecological momentary assessment design. Over a four-week period, we asked participants ($N=52$; 67% Female; 80% White) to report their levels of feeling loved and expressing love six times a day. Using a continuous-time process model, we explored individual differences in inertia (i.e., persistence of a process over time) and cross-influences of felt and expressed love over time. We found that increases in expressing love led to increased feelings of being loved over time; however, increases in felt love did not lead to increases in expressing love. Notably, participants who experienced more persistent feelings of love (that is, greater inertia over time) indicated higher levels of flourishing. These results suggest new avenues for psychological well-being interventions which target increasing loving feelings through encouraging more expressions of love.

## Introduction

From romantic partnerships to friendships and familial bonds, love weaves through the fabric of our daily lives. In the scientific literature, love has been explored in both romantic and non-romantic contexts. Across these, multiple forms of love have been distinguished, with the primary types being passionate, companionate love, and compassionate love (see, e.g., [1–3]). *Passionate* love can be broadly described as a longing or desire to be in another's presence in a romantic context [4]. Experiences in daily life of passionate love can include spending time with a significant other, a desire for togetherness when not in their presence [5], or physical expressions of love such as hugging or kissing [6]. It can also include physiological responses such as

**Data availability statement:** All data and analysis files are available from the Open Science Framework database at https://osf.io/gfsxw/

**Funding:** Support for this work was provided by grant #48192 from the John Templeton Foundation (ZO, SH, JV) and the Edna Bennett Pierce Endowed Chair in Caring and Compassion at Penn State (RR). The sponsors or funders did not play any role in the study design, data collection and analysis, decision to publish, or preparation of the manuscript. https://www.templeton.org/ https://prevention.psu.edu/peace/people/

**Competing interests:** The authors have declared that no competing interests exist.

an increased heart rate when thinking about the person or even a feeling of euphoria when requited, reflecting its deeply affective and often transient nature. In contrast, *companionate* love, also known as attachment, represents a less intense, yet more enduring bond, characterized by a warm, steady, and dependable feeling towards another, built on a foundation of trust, intimacy, and commitment [1,5]. Companionate love could exist in non-romantic contexts as well, such as between friends or family members [7]. *Compassionate* love (e.g., [8,9]) is characterized by a selfless concern for another's well-being when they are in need and a motivation to act to alleviate their suffering, in both romantic and non-romantic contexts. Compassionate love is often manifested through acts of support, which can be instrumental support in the form of acts of service or emotional support in the form of words of encouragement, understanding, kindness, or a thoughtful gift.

Across these contexts and forms, love has been described as a multifaceted experience that integrates biological, social, behavioral, cognitive and emotional aspects [3,10]. For example, biological markers of love may include changes in brain activity or fluctuations in hormones or neurotransmitters [4,10,11]. Social and behavioral components are reflected in actions such as spending quality time with loved ones or engaging in kind gestures [10]. Meanwhile, the cognitive part encompasses the mental processes involved in considering another's positive qualities (see, e.g., [4]) or the knowledge of the cultural agreement on expressions of love (see., e.g., [12]). These various facets interrelate to form the complex construct of love.

Several studies have placed strong emphasis on the emotional aspects of love, which have sparked longstanding debates. Research into the emotional aspects of love dates back to 1884 [13]; however, it was not until the 1970s that the emotion-focused conceptualization started to gain traction [5], as there has been disagreement in academic circles about whether love can be viewed an emotion. Shaver and colleagues [13] explained that some scholars do not conceptualize love as an emotion due to its dependence on a connection with another entity, while others note that the enduring nature of love, such as in relationships, makes it problematic to be considered an emotion, as emotions are more short-lived experiences. Additionally, Berscheid [1] also noted that there are differing views among scholars regarding the definition of an emotion, making it difficult to then determine if love qualifies as an emotion. Despite these different viewpoints, exploring love as an emotion has emerged as an active field of research, highlighting the important features of love when conceptualized as an emotional state. For instance, love has been studied as a temporally dynamic burst of emotion shared between individuals, leading some researchers to regard love as a "supreme emotion" [14,15].

Recent studies have also started to characterize how love is experienced in daily life and its contribution to well-being (see, e.g., [12,14,16,17]). The theory of "positivity resonance" [14] emphasizes the emotional experience of love, while also describing its unique physiological and behavioral aspects. It suggests that love in everyday life emerges from small acts or gestures creating positivity resonance, that foster positive connections between individuals. This line of research situates love within the context of everyday interactions, termed "love in everyday life" (LEL), which

includes but is not limited to romantic settings (see, e.g., [16–20]). Heshmati and colleagues [12] also showed that people in the US are in agreement on the experiences that elicit love in daily life, spanning from moments of deep connection with loved ones to interactions with strangers. In other words, experiences of love can occur between people across different "social ties" [21]. In line with this, when asking non-experts to provide examples of love, also known as a prototype approach, they often recount moments experienced with family members, friends, significant others, and even strangers [2,12,22]. Experiences of love in everyday life have been linked with psychological well-being and improved sleep quality across the lifespan [15–17,19,20,23,24]. However, the daily dynamics between feeling and expressing love and their implications for well-being are largely unexplored.

Consistent with research on love and loving-kindness (e.g., [25,26]), we distinguish between two aspects of LEL: a felt aspect, that captures how much a person feels loved when receiving love from someone, and an extending aspect, that focuses on the behavior of a person expressing loving feelings towards someone. We label the felt aspect of love as "feeling loved" or "felt love," and the extending aspect of love as "expressing love" or "expressed love." Research has explored giving and receiving communal support and responsiveness within the context of existing relationships (e.g., [27,28]). Lemay et al. [28] described the link between an individual's perception of support from their partner and their own extended responsiveness. In this context, the authors explored the mechanism through which extended and perceived support are related, with individuals projecting their own support onto their perceptions of their partner [28]. While the study conducted by Lemay and colleagues was contextualized to support within a romantic partnership, and noted to be limited to a dyadic interaction [28], this mechanism can be extended to the context of LEL, suggesting a possible linkage between expressing love and feeling loved. Consistent with research on the cultivation of another form of love, compassion (e.g., [29]), it is possible that felt and expressed love dynamically influence each other, creating a positive feedback loop where an increase in expressed love yields an increase in felt love over time. Our study aims to capture the dynamics of these two aspects of LEL as they unfold over time.

Our conceptual model for love dynamics is similar to that of affect dynamics [30]. We assumed that the two aspects of love – felt and expressed – can be described in terms of their baseline levels, intra-individual (stochastic) variability, and inertia (e.g., time to return to baseline following changes in the system or auto-correlation), and most importantly, their possible asymmetric influence on each other (cross-influences) over time. Specifically, in our study we used a continuous-time stochastic process model to capture the most important dynamical features and interactions of feeling and expressing love over time.

In what follows, we focus on examining love dynamics in terms of inertias and cross-influences. Inertia describes how a process persists over time, as opposed to being regulated back to its baseline [31], and its level describes the persistence of momentary states over time. Higher inertia means that the level of loving feelings changes more slowly, while lower inertia refers to a quick regulation back to baseline. Each person can be described with their own inertia estimates of felt and expressed love, and these two can differ from each other. In addition, cross-influences (i.e., cross-effects) capture the reciprocal relationship between felt and expressed love (and vice versa) over time. The magnitudes of these cross-influences express how likely it is that changes in felt love/expressed love are followed by similar changes in the expressed love/felt love over time. These cross-influences are also individual-specific and can be asymmetrical. Therefore, in our study we captured how likely it was that changes in the intensity of feeling loved were followed by changes in the intensity of expressing love, as well as the opposite, for every individual. We also derived group-level estimates for these inertias and the cross-influences via multilevel modeling.

To explore the links between sources of individual differences in love dynamics and psychological well-being, we selected relevant trait-level measures, such as general happiness, emotional well-being, and flourishing. While these three scales all explore dimensions of mental health and happiness, they each provide a unique aspect which may be informative in the context of LEL. General happiness provides comparative information [32], while emotional well-being captures mental health more wholly [33]. The flourishing scale includes additional dimensions regarding relationships,

meaning, and general attitude toward the future [34]. In fact, Major et al. [20] found a link between flourishing mental health and positivity resonance. In relation to love dynamics in everyday life, each scale provides a component which may uniquely influence felt and expressed love. Biological sex was also included in the analysis as prior studies suggested possible gender differences [35].

## Methods

### Recruitment and participants

We recruited 52 adults for a 28-day study in a northeastern college town. Of the participants, approximately 80% indicated White or Caucasian, 10% indicated Asian or Pacific Islander, 6% indicated Hispanic or Latino, and 4% indicated Black or African American. For biological sex, 67% indicated female and 33% indicated male. The participants ranged from 19 to 65 years of age ($M = 30.19$, $SD = 10.14$). Out of the 52 participants, 46 indicated they were in a relationship, while 6 indicated being single. For level of education, 19 participants indicated college degree or below, 19 participants indicated a bachelor's degree, and 14 indicated a master's degree or above. No participants were excluded from the analysis. All participants provided written informed consent. The study was approved by Pennsylvania State University's Institutional Review Board (STUDY1017) and was performed in accordance with relevant guidelines and regulations. The study design and analyses were not preregistered.

### Procedure

Data analyzed here was part of a larger experience sampling study that explored people's psychological well-being. We used a subset of the variables from this parent study needed to answer our research questions. Full details on the parent study with all measures are available via https://osf.io/7hvce/; relevant details are provided below.

For calculating correlations between individual-specific dynamics and the trait-level covariates (e.g., emotional well-being scores), power analysis showed that we needed a minimum of 46 people for detecting medium sized ($r = .4$) effects. Our sample size of N = 52 was sufficient for that. More importantly, note that the Bayes factor approach we used for testing for correlations allows for quantification of evidence in the data in a statistically principled manner for any effect size, and without requiring the binary decision of significant vs. non-significant. The trait-level variables were measured at the beginning of the study during an in-person intake session.

After the intake session, participants were asked to complete self-report surveys (that included the two items on their love experiences) on their own phones at random times (maximum of 6 times a day) during their regular waking hours for 28 days, during their daily life. The choice of the sampling frequency was based on previously published studies that indicated that the selected continuous-time longitudinal model requires approximately 60 measurements per person (see, e.g., [36]). The average number of observations for a person in our study was 157 ($SD = 15$).

### Measures

Felt love was measured repeatedly with the item "How much do you feel loved right now?" and expressed love was measured by the item "Since the last survey, I have been expressing love." Individuals responded on a sliding scale from 0 to 100, and a label of "Not at all" was shown above the left side of the scale, corresponding to 0, while "Extremely" was displayed above the right side of the scale, corresponding to 100. It should be noted that responses to each item were not necessarily in relation to the same loving connection. Also, previous work by Heshmati et al. [12] and Dickens et al. [19] has shown that the layperson's understanding of loving scenarios in everyday life do exist outside of the context of romantic relationships. Based on these findings in the context of this study, it is reasonable to assert that the responses provided by the participants are largely in alignment with the cultural consensus of love occurring in everyday life scenarios.

Biological sex was coded as a binary variable (*Male* = 0, *Female* = 1). General happiness was measured using the Subjective Happiness Scale [32], consisting of four statements with response values ranging from 1 (not very happy) to 7 (very happy); the mean score was used for the analysis. The Flourishing Scale [34] was used to measure flourishing and asked participants for their level of agreement for each item on a scale of 1–7 for eight items; the sum score was used for the analysis. Emotional well-being was measured using the Short-Form Health Survey [33] and consisted of four questions with values ranging 1 (all of the time) to 6 (none of the time); items were recoded as needed so higher values indicated higher well-being and the mean value was used for the analysis. Descriptive statistics, including the mean and standard deviations of the trait level measures, are shown in Table 1. The mean levels of general happiness, flourishing, and emotional well-being were aligned with average levels for the scales in similar populations (see, e.g., [32] for general happiness, [34] for flourishing, and [37] for emotional well-being). Population averages are not established for felt and expressed love—however, they were both measured on a scale from 0 to 100, and their mean values reported in Table 1 were higher than what we could consider the middle point given the scale (i.e., 50), suggesting that our participants were more on the higher end of feeling and expressing love.

## Data analysis

We captured the time dynamics of love with a continuous-time stochastic model in which we allowed individuals to differ in their dynamical characteristics. While previous studies captured positivity resonance in discrete time [15], a continuous-time stochastic model allowed us to capture dynamical features as an evolving system in high-intensity, irregularly-spaced ecological momentary assessment (EMA) data [38]. Several studies have used a similar continuous-time stochastic process model to account for dynamical changes in core affect [30,39] and well-being [40] in daily life.

We measured participant $p$'s felt and expressed love levels at time $t$ via self-reports, denoted by the bivariate vector $\boldsymbol{y}_p(t)$. We assumed that these measures were perturbed with measurement noise, and that love dynamics evolved on a latent level, and specified our measurement model as $\boldsymbol{y}_p(t) = \boldsymbol{\eta}_p(t) + \boldsymbol{\tau}_p + \boldsymbol{\epsilon}_p(t)$, with the 2-by-1 vector $\boldsymbol{\eta}_p(t)$ indicating latent levels of felt and expressed love, a 2-by-1 vector $\boldsymbol{\tau}_p$ capturing the manifest intercept, and the 2-by-1 vector $\boldsymbol{\epsilon}_p(t)$

**Table 1. Descriptive statistics of the variables in the study.**

| Variable | N | % | Mean | SD |
|---|---|---|---|---|
| Sex: Male | 17 | 33 | | |
| Sex: Female | 35 | 67 | | |
| Relationship Status: In a relationship | 46 | 88 | | |
| Relationship Status: Single | 6 | 12 | | |
| Education: College degree or below | 19 | 36 | | |
| Education: Bachelor's degree | 19 | 36 | | |
| Education: Master's degree or above | 14 | 27 | | |
| General Happiness | | | 5.22 | 1.43 |
| Flourishing | | | 45.44 | 8.85 |
| Emotional Well-being | | | 4.72 | 0.85 |
| Felt Love | | | 68.44 | 15.23 |
| Expressed Love | | | 64.62 | 17.63 |

Note. Number of participants and corresponding percentages are shown for the Sex, Relationship Status, and Education variables. For the continuously scaled variables, the Mean column shows the average across individuals, the Standard Deviation (SD) column displays the corresponding between-person variability for each variable. For the EMA measures of felt and expressed love, the within-person mean was calculated for each person – the mean and standard deviation of these are displayed.

quantifying the manifest residuals (see more details in [41]). Changes over time in the latent process, $d\boldsymbol{\eta}_p(t)$, were modeled through a bivariate stochastic differential equation model, specified as:

$$d\boldsymbol{\eta}_p(t) = \left(\boldsymbol{A}_p\boldsymbol{\eta}_p(t) + \boldsymbol{b}_p\right) dt + \sum_p d\boldsymbol{W}(t)$$

where $\boldsymbol{A}_p$ is the person-specific 2-by-2 drift matrix, $\boldsymbol{b}_p$ is the person-specific continuous-time intercept vector, $\boldsymbol{\Sigma}_p$ is the person-specific 2-by-2 diffusion coefficient matrix, and $\boldsymbol{W}$ is a Brownian motion process. This model captures the continuous-time dynamics through the drift matrix, $\boldsymbol{A}_p$, which is composed of the autocorrelations (inertia) on its diagonal and cross effects for felt and expressed love on its off-diagonal. In other words, it was this drift matrix that captured how felt and expressed love were linked to their respective past values as well as the cross-influences over time. All dynamic features (i.e., inertia and cross effects) were allowed to differ across individuals – that is each individual's felt and expressed love dynamics were modeled as their own evolving system. To transform the diagonals of the drift matrix into inertia, a matrix exponential was applied. We fit this continuous-time stochastic model in a multilevel Bayesian framework to provide an intuitive understanding of how the love dynamics evolve both at the individual and group level.

For data analysis, we used the R [42] package ctsem [41]. We ran two chains with 3,000 iterations, with default, weakly informative priors distributions. Additionally, most parameters estimated varied by person and initial time data points were not included as the expressed love question refers to the previous survey which would not be applicable for the initial data point. Data points which occurred less than one minute after a previous data point were excluded.

Individual-specific estimates of these love dynamics were tested for associations of the following trait level variables: general happiness, biological sex, emotional well-being, and flourishing, to explain the link between relatively stable person characteristics and love dynamics. We used JASP [43] to calculate correlations and corresponding Bayes factors. We chose the Bayes factor as it allows for quantification of evidence in the data in a statistically principled manner for any effect size (without requiring the binary decision of significant vs. non-significant). Data and analysis scripts used for this study are available on the Open Science Framework website at the following link: https://osf.io/gfsxw/.

## Results

The group-level estimates of key dynamical parameters (i.e., elements of the drift matrix) and their 95% credible intervals are shown in Table 2. The credible interval is interpreted as the range of numbers containing the true parameter with 95% probability (given the data and our model specifications; [44]). Based on these estimates, we illustrated love dynamics over time (in hours) in Fig 1 using the R [42] packages ctsem [41] and ggplot [45]. The curves are based on the mean estimates in Table 2, shaded areas display the 95% credible intervals. Fig 1 shows group-level love dynamics over time in terms of inertias and cross-influences of felt and expressed love.

The 95% credible intervals of the group-level drift matrix elements – that quantify felt and expressed love inertias – did not contain 0, indicating these were credibly different from 0. In the left panel of Fig 1, the felt and expressed love inertias

**Table 2. Group-level love dynamics estimates.**

| Time dynamics features | Mean | 95% CI |
|---|---|---|
| Felt love inertia | −0.065 | [−0.130, −0.014] |
| Expressed love inertia | −0.900 | [−1.530, −0.512] |
| Expressed to felt love cross-influence | 0.820 | [0.358, 1.578] |
| Felt to expressed love cross-influence | −0.038 | [−0.088, −0.005] |

Note. Group-level mean estimates of the time dynamics features of love are shown in the second column with their corresponding 95% credible intervals in the third column.

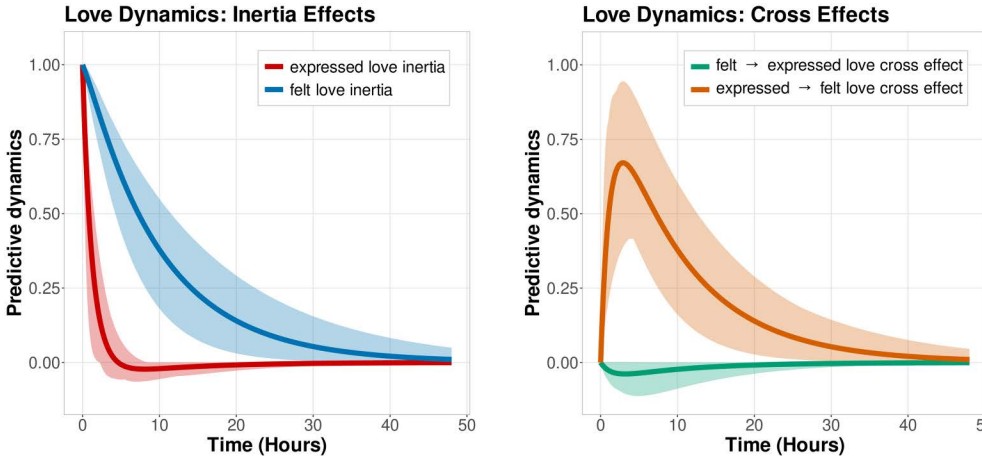

**Fig 1. Illustration of the felt end expressed love dynamic estimates.** In the left panel, the cross-influences and inertias for felt and expressed love are shown based on the group-level estimates from Table 2, with shaded areas displaying the corresponding 95% credible intervals. Expressed love inertia and felt love inertias are displayed in red and blue, respectively. In the right panel, the expressed to felt love cross-influence is shown in orange, the felt to expressed love crossed effect in green.

over time are shown in blue and red, respectively. Examining the time axis (horizontal) allows us to explore the duration of loving feelings across different time periods. For example, around four hours, felt love inertia was around 0.75, as can be read off on the predictive dynamics axis (vertical). That value indicates the extent to which the current level of felt love is predicted by the level of felt love four hours ago. A decay in inertia over time was expected based on literature on emotion processes [46].

Fig 1 in the right panel illustrates the cross-influences from felt to expressed love and vice versa, corresponding to the estimates of the last two lines in Table 2. Neither of these credible intervals contained 0, indicating that extending and felt love dynamics were credibly linked to each other across time. Similar to the panel on the left, the right panel illustrates the values of the cross-influences on the vertical axis across many durations of time on the horizontal time axis. For example, the cross-influence for expressed love to felt love (shown in orange) remained positive throughout the interval shown on the graph, peaking around three hours with a value around 0.64. Conceptually, this means that the more intensely participants expressed love, the stronger they were feeling loved, with this dynamic peaking at a lag of three hours. Felt love to expressed love dynamics showed a different pattern. We found that increases in feeling loved were followed by slight decreases in expressing love – as shown by the green curve in the right panel of Fig 1. This effect was very small, with the limits of the 95% credible interval approaching 0 after only a few hours. This finding suggested that when people felt loved, they afterward became less likely to be expressing love – however, these dynamics were not very pronounced (i.e., negligible effect size).

To capture individual differences in felt and expressed love dynamics, dynamical features were allowed to differ across individuals, and we studied their links with trait-level variables. Results are shown in Table 3. We included Pearson's correlation values (first line in each cell) with the evidence that the correlation is different from 0, summarized by the Bayes factor (second line in squared brackets). The higher the Bayes factor, the more evidence there was that a correlation differed from 0. Generally, a value between 1 and 3 indicates anecdotal evidence of a non-zero correlation, a value between 3 and 10 indicates substantial evidence for it, while anything above 10 indicates strong evidence [47].

There was strong evidence for positive correlation between flourishing and felt love inertia. Higher levels of flourishing were linked to slower changes in feeling loved (higher inertia), suggesting people with higher levels of flourishing would have longer-lasting loving feelings. Aside from this correlation, we found no additional evidence for linkages between the

**Table 3. Correlations and corresponding Bayes Factors for the love dynamics individual-level correlation estimates with the trait level variables.**

| Time dynamics features | Sex | General Happiness | Flourishing | Emotional Well-being |
|---|---|---|---|---|
| Felt love inertia | −0.060 [0.189] | 0.262 [0.956] | 0.411* [14.778] | 0.192 [0.426] |
| Expressed love inertia | −0.176 [0.369] | −0.051 [0.184] | −0.060 [0.189] | −0.110 [0.231] |
| Expressed to felt love effect | 0.313 [2.084] | 0.303 [1.769] | 0.194 [0.435] | 0.203 [0.477] |
| Felt to expressed love effect | 0.148 [0.294] | 0.329 [2.714] | 0.262 [0.960] | 0.187 [0.407] |

Note. Asterisk indicates a Bayes factor larger than 10.

inertias and our selected trait level measures. With respect to the cross-influences, there was some weak evidence for an association between general happiness and the felt love to expressed love dynamic, suggesting people who tended to be happier also expressed more love when their own feelings of love increased.

## Discussion

In our study we assumed that love in everyday life involved both the feelings of love and the affection we show to others, and that these aspects of love would continuously change as time goes on. We conceptualized feeling and expressing love in daily life as a dynamical system unfolding over time and estimated its properties with data from an EMA study. In our findings, the expressed to felt love cross-influence provided evidence of the dynamic interplay between these aspects of love. Results showed that the cross-influences of expressed and felt love were credibly linked, indicating that increases in expressed love levels were followed by increases in felt love levels over time, peaking at three hours after the expression of love and decaying after this point. One possible explanation for this finding is that by expressing more love, an individual may also receive more expressions of love in return, resulting in more felt love. This result implies that expressing love may be a potential route to increase felt love. Expanding on this idea, we propose that LEL could be conceptualized as a skill that can be developed. With regular practice and heightened awareness in daily activities (see, e.g., in [48]), individuals can become more adept at receiving and expressing love, which in turn can cultivate stronger feelings of love.

Additionally, the cross-influence from felt love to expressed love was found to be small, practically negligible – however this cross-influence was in the opposite direction, meaning that as felt love was increasing, expressed love was decreasing. These results may indicate that the lasting effect of felt love could be interfering with expressing love, explaining why felt love was not highly linked to expressed love over time. Conceptually, these results align with the idea of self-cherishing possibly impeding an individual's expressed love. If an individual is self-cherishing and preoccupied with their own felt love, this could stall additional acts of expressed love [49,50].

Although the context of love for this study was broader, these findings are similar to the Lemay et al. [28] findings which suggested that an individual projects their own extended support on their perception of their perceived support from their partner. In the context of this study which measures love as it occurs in daily life, across any and all relationship contexts, this could be compared with the cross-influence of expressed love to felt love and understood to be a similar mechanism.

We found that the feeling of being loved endured over time – felt love inertia was high, with levels showing strong correlation even after eight hours. Expressed love inertia states were moderately associated after one hour, but their connection practically disappeared in the next couple of hours. It is possible that felt love has an appraisal aspect that leads to a more enduring subjective feeling state, while expressed love is a motivated behavioral state involving the wish or feeling for another to be happy. It may be that the subjective feeling state is more persistent than the motivated behavioral state,

though we are aware of little research on these dynamics. This represents an interesting area for future research regarding the dynamics of love in a variety of relationships – attachment relationships, romantic relationships, or interactions with strangers. Future studies might also explore how felt love inertia relates to "savoring" which involves mindfully evoking emotions from positive experiences [51]. Perhaps savoring moments of everyday love is the quintessential example of the construct of savoring. Also, when compared to other psychological phenomena modeled through the same dynamical approach in daily life studies, both felt love and expressed love inertia was considerably higher than inertia for valence and arousal [52], and positive and negative affect [53], suggesting that these love experiences are more resistant to change.

From an evolutionary perspective, it stands to reason that humans should be predisposed towards behaviors that are beneficial for their survival. In his evolutionary theory of love, Buss [54] described love as a multifaceted experience composed of emotion, cognition, and behavior. He posited that love increases the likelihood of survival through mechanisms such as commitment, resource sharing, and caring for offspring. For a cooperative and interdependent species, the sharing of positive emotions like kindness, compassion and non-romantic love provides a metaphorical glue for healthy social relations and the development of trust, as well as supports functioning and survival at the group level and the level of the individual (e.g., [55]). Other-regarding emotions may have evolved as an essential dimension of both group cooperation and attachment behavior and, thus, love in everyday life may have evolved outside of romantic relationships as a key means of survival at both the group and individual levels of analysis (e.g., [56]). In the context of our study, expressing love can be seen as an evolutionarily desirable behavior, which helps generate loving feelings as well as cultivate and solidify connections with others. Our findings, which indicate a connection between greater felt love inertia and increased well-being, coupled with other research that associates higher levels of felt love with improved well-being (e.g., [16]), suggest that these dynamics of love may be beneficial for survival. Note, however, that our data do not allow us to draw any conclusions about the potential heritability of these dynamical parameters.

Future directions include interventions aimed at increasing an individual's expressed love or flourishing as these mechanisms could also increase felt love for the individual [57]. They also may test if strategies for the maintenance of loving feelings (i.e., by encouraging the expression of loving behaviors) would lead to more flourishing (e.g., [29,57]). While the current finding on the associations between higher levels of well-being with more persistent feelings of love (i.e., higher inertia) does not allow any conclusions to be drawn about causality, the theory of "broaden-and-build" [58] proposes potential mechanisms through which this can happen. In this theory, experiences of positive emotions can empower individuals to broaden so that they can access additional mental, physical, or cognitive resources which enable more positive outcomes [58]. Further studies are needed to test whether increasing loving feelings would in turn elicit changes in well-being.

Additionally, we acknowledge that this study does not parse out the occurrence of interactions between the participants and others. In other words, we did not capture if an individual had an experience for example with a significant other (stronger social tie) versus a stranger (weaker social tie) when they reported their levels of felt or expressed love. It is possible that interactions with particular individuals may have a more lasting effect depending on the quality or quantity of love exchanged in an interaction. Future studies might also explore the differences between individuals in their living arrangements or frequency of interactions across different types of bonds or relationships.

## Limitations

Limitations include the predominantly White or Caucasian sample from northeastern US. As previously stated, the sample size of 52 participants was sufficient to detect medium sized effects, but it was not large enough to detect small sized effects. We also did not collect data from individuals the participants interacted with throughout the study or ask the participants to report expressions of love towards them; therefore, we were not able to capture expressed love from others towards the participants, enabling us to discern the pathways through which felt love was increased. Additionally,

we utilized single-item measures for felt and expressed love which may not capture some of the complexities when conceptualizing love as an emotion (see, e.g., [59]). Future research is needed to study possible racial, ethnic, and regional differences.

## Conclusion

This research contributes to the research literature on love by examining the dynamical interplay between different aspects love across time. Our findings showed that the feelings of being loved can be maintained over time, and the participants who reported more enduring feelings of love also had higher levels of flourishing. Additionally, the participants who frequently expressed love also experienced an increase in feelings of love as time progressed. Considering these felt and expressed love dynamics, complemented with techniques for cultivating LEL through reflection and practice [25], it seems possible to facilitate the development of psychological well-being interventions. Importantly, this research unlocks the opportunity for interventions to target increasing expressed love to increase an individual's feelings of love over time.

## Author contributions

**Conceptualization:** Lindy Williams, Saida Heshmati, Joachim Vandekerckhove, Robert W. Roeser, Zita Oravecz.

**Data curation:** Lindy Williams, Saida Heshmati, Joachim Vandekerckhove, Zita Oravecz.

**Formal analysis:** Lindy Williams, Yanling Li, Joachim Vandekerckhove, Zita Oravecz.

**Funding acquisition:** Joachim Vandekerckhove, Zita Oravecz.

**Investigation:** Lindy Williams, Robert W. Roeser.

**Methodology:** Lindy Williams, Sharon H. Kim, Yanling Li, Saida Heshmati, Joachim Vandekerckhove, Zita Oravecz.

**Software:** Lindy Williams, Yanling Li, Zita Oravecz.

**Writing – original draft:** Lindy Williams, Sharon H. Kim, Yanling Li, Saida Heshmati, Joachim Vandekerckhove, Robert W. Roeser, Zita Oravecz.

**Writing – review & editing:** Lindy Williams, Zita Oravecz.

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
