## [Decision Letter · Decision Letter 0]

Mar 30 2025

Dear Dr. Oravecz,

Thank you for submitting your manuscript to PLOS ONE. After careful consideration, we feel that it has merit but does not fully meet PLOS ONE’s publication criteria as it currently stands. Therefore, we invite you to submit a revised version of the manuscript that addresses the points raised during the review process.

We look forward to receiving your revised manuscript.

Kind regards,

Marta Kowal

Academic Editor

PLOS ONE

Journal Requirements:

**Additional Editor Comments:**

Dear Authors,

Based on the reviewes and my reading of your work, I would like to invite you to resubmit your manuscript with minor changes.

Kind regards,

Marta Kowal

Reviewers' comments:

Reviewer's Responses to Questions

**Comments to the Author**

1. Is the manuscript technically sound, and do the data support the conclusions?

Reviewer #1: Yes

Reviewer #2: Partly

Reviewer #3: Yes

2. Has the statistical analysis been performed appropriately and rigorously?

Reviewer #1: I Don't Know

Reviewer #2: I Don't Know

Reviewer #3: Yes

3. Have the authors made all data underlying the findings in their manuscript fully available?

Reviewer #1: Yes

Reviewer #2: Yes

Reviewer #3: Yes

4. Is the manuscript presented in an intelligible fashion and written in standard English?

Reviewer #1: Yes

Reviewer #2: Yes

Reviewer #3: Yes

Reviewer #1: Thank you for the opportunity to review this manuscript. As someone who researches love in romantic relationships, I was drawn to the paper from the moment I read the abstract. I looked forward to learning a bit.

Please note, however, my expertise does not extend to the methods employed in this study and therefore I am not able to comment on the analysis. I have informed the editor of this and suggested that they seek a second reviewer who has expertise in the methods you employ.

This study investigated two aspects of love (felt love and expressed love) in North American adults using EMA. If I understand correctly, the authors are particularly interested in inertia, which is a proxy for some sort of causal relationships between the two components of love, but also measures of wellbeing. Overall, I think the study is novel, interesting, and reasonably well presented. Nonetheless, there are a number of issues that I have identified with the manuscript that I think warrant either amendment or, at the very least, consideration by the authors.

Please note, I have included some references below. I am not suggesting that you cite these, they are just for your consideration and a starting point…

Intended audience

I wonder about the intended audience of the manuscript. As a romantic love researcher, I think the study is very interesting and has the potential to influence researchers’ thinking on the topic. If you are wanting to contribute meaningfully to the literature on love in romantic relationships (which you may not want to), you may think about developing your introduction and discussion. The introduction could better situate the current study within the extant literature on love (the largest area of love research is love in romantic relationships). This may involve a brief summary of the different types of love, including in romantic relationships (e.g., passionate/ romantic love and companionate love – see {Walster, 1978 #557}. This could also suggest a bit more in terms of future research. You jump into the theory of positivity resonance a bit quickly in the introduction, and I think you could situate the reader a bit better. See {Machin, 2022 #1300} or {Fehr, 1991 #1846} for the various types of love.

Conceptualising love as an emotion

I also think it is narrow to describe love as a “supreme emotion.” While some researchers still apparently believe this, most contemporary researchers studying love in romantic relationships recognise love is a constellation of psychological features, including emotions, cognitions, and behaviours ({Bode, 2021 #2276} {Sternberg, 1986 #565}. I appreciate that some of the authors specialise in emotions, but this is narrowly focused and ignores the overwhelming body of evidence. Perhaps soften the language a bit to indicate that focussing on the emotional aspect of love is a valid and necessary approach. Emotion is one component of love… The authors, themselves, recognise that love is associated with “small acts or gestures.”

Lack of acknowledgement about the influence of different relationships

The discussion does not really detail how different types of relationships may influence the results. For example, people who live with a romantic partner or in a family situation may experience both felt and expressed love more than a single person simply because of exposure to a loved one. You may want to include acknowledgement of this, and this should be suggestive of future research. Different types of relationships may illicit different felt and expressed love, and different inertia and wellbeing.

Lack of overarching theory

I think your lack of overarching theory is a major weakness of the paper.

You refer to some theories, but these are not overarching theories. They are more mechanistic (and to be honest, are not very explanatory). I refer you to the following, by some insightful researchers:

Many subfields within psychology lack any overarching, integrative general theoretical framework that would allow researchers to derive specific predictions from more general premises. Without a general theoretical framework, results are neither expected nor unexpected based on how they fit into the general theory and have no implications for what we expect in other domains. This situation is thrown into stark relief by comparing psychology textbooks with those in other sciences. Rather than building up principles that flow from overarching theoretical frameworks, psychology textbooks are largely a potpourri of disconnected empirical findings on topics that have been popular at some point in the discipline’s history, and clustered based on largely American and European folk categories. Outside of psychology, useful theoretical frameworks tell scientists not only what to expect, but also what not to expect (Muthukrishna & Henrich, 2019) p. 221).

This is the case with your manuscript. I suggest you see if the theory of evolution may be able to make better sense of your study and findings. This could be easily done by indicating what the functions of the inertia or wellbeing are in love. Why would this inertia or wellbeing have benefited reproduction or survival – once we know that, we know why it works this way. Perhaps inertia exists because love is a reciprocal process, whereby the purpose is to bond two entities together. Perhaps feeling loved is the rewarding aspect of expressed love??? Perhaps wellbeing is a way of saying that love is good for us??? I refer you to {Buss, 2019 #124}. If you don’t feel comfortable with evolution, you may be able to identify alternative theories to explain your findings. There might be a social theory that could do the job.

Ideas for future research

I think you could do better with ideas for future research. This study opens up all sorts of questions. Are there sex differences. Are their age differences. Are there differences based on relationship status. Does felt or expressed love affect behavior? Many more I am sure.

Little things

If you can, describe the participants in more detail (see {Bode, 2023 #2264}).

Is there some sort of power analysis/ sensitivity analysis that can be conducted for this type of analysis? If so, add that.

“Male” and “female” are biological sexes, not genders.

Please indicate what 1 and 7 mean in the Flourishing Scale as well as 1 and 6 in the SFHS and 1 and 7 in the SHS.

Can you provide any context to the descriptive statistics. Are the means high, low, expected???

I don’t really care, but APA format says you don’t just repeat stats in text and tables…

Figure 1 is pretty hard to make sense of. Perhaps describe it in text a bit more.

Your choice of the term “expressed” love is misleading/ confusing. Can you come up with a better term. Expression is suggestive of behaviour or felt love could be a form of expression.

“These findings cast LEL as a dynamic flow of felt and expressed love which unfolds over time.” This is not a useful description. I don’t know what it means.

Unless the journal requires it, don’t have the heading Constraints of generality. See ideas for expressing issues with generalisability here: {Bode, 2023 #2264}

You need a limitations paragraph, which should include generalisability issues

Discuss causality in more detail (inertia and wellbeing) - what are the potential mechanisms

Bode, A., & Kowal, M. (2023). Toward consistent reporting of sample characteristics in studies investigating the biological mechanisms of romantic love. Front Psychol, 14, 983419. doi:10.3389/fpsyg.2023.983419

Bode, A., & Kushnick, G. (2021). Proximate and Ultimate Perspectives on Romantic Love. Front Psychol, 12, 573123. doi:10.3389/fpsyg.2021.573123

Buss, D. M. (2019). The evolution of love in humans. In R. J. Sternberg & K. Sternberg (Eds.), The New Psychology of Love (Second ed.). Cambridge, UK: Cambridge University Press.

Fehr, B., & Russell, J. (1991). The Concept of Love Viewed From a Prototype Perspective. J Pers Soc Psychol, 60(3), 425-438. Retrieved from http://ovidsp.ovid.com/ovidweb.cgi?T=JS&PAGE=reference&D=ovfta&NEWS=N&AN=00005205-199103000-00010

Machin, A. (2022). Why we love. London: Weidenfeld & Nicolson.

Muthukrishna, M., & Henrich, J. (2019). A problem in theory. Nature Human Behaviour, 3(3), 221-229. doi:10.1038/s41562-018-0522-1

Sternberg, R. J. (1986). A triangular theory of love. Psychological Review, 93(2), 119-135. doi:10.1037/0033-295x.93.2.119

Walster, E. H., & Walster, G. W. (1978). A new look at love. Reading, MA: Addison-Wesley.

Reviewer #2: I appreciate having had the opportunity to review the authors’ article, entitled How much we express love predicts how much we feel loved in daily life. Because I am not an expert on the quantitative approaches utilized by the authors, I have made the theoretical contributions of the paper the primary focus of my review (with permission from the editor). Overall, I felt that the findings were very intriguing. However, I felt that the authors’ interpretations of the work required substantial revision.

• The authors consistently refer to expressed and felt love as dimensions of love experiences. In the discussion, the authors also describe love experiences in everyday life as a skill. However, although one may motivate the other, it’s not clear how both could be considered dimensions of the experience of love. As an analogy, one would not state that felt anger and behavioral aggression are dimensions of the experience of anger. Although the feeling of anger and the motivation to aggress when angry could be outputs of an anger program in the mind, it’s not clear how these are dimensions of the experience of anger; only the feeling state associated with anger (and various other, related phenomena—such as a tightening of chest, an increase in heart-rate, etc.) characterize the experience of anger. The introduction should be amended to remove this point or clarify the authors’ meaning.

• The authors write that “on average, felt love decayed at a slower rate than the expressed love, suggesting a stronger lingering effect of felt love than expressing love” and discuss this in the General Discussion. However, in addition to not representing two dimensions of love experiences, it's not clear that these phenomena are directly comparable in the first place. Felt love is an affective state, whereas expressed love is an action. Comparing the decay of one to the decay of the other is analogous to comparing decays in acts of aggression to decays in internal feelings of anger; they’re completely different phenomena. Thus, it’s not clear how one could interpret this comparison.

One alternative possibility is to compare felt love and expressed love to findings from other published research. For instance, if felt love/expressed love decayed at a slower or faster rate than has been suggested in previous research on felt hatred/expressed hatred, this could be very interesting. If no such comparisons are available in the literature (or if such comparisons are inappropriate in the opinion of the authors), this is not needed. However, in either case, any comparisons between felt and expressed love should be removed or explicitly justified in the manuscript.

• The authors mention projection as a potential mechanism for the relationship between felt and experienced love, but other interpretations are possible. In particular, expressions of love could lead to others expressing more love in return, and this reciprocity could increase felt love. This possibility should be discussed by the authors.

• The limitations section of the General Discussion is extremely short and omits a number of key issues, such as the small sample size, ambiguities in the interpretation (described above), the use of single-item measures, the absence of an item measuring others’ expressions of love toward the participant (e.g., one could feel unloved despite others attempting to express love—just as one could feel lonely despite interacting with others), and so on. These limitations should be addressed in the limitations section.

Other points:

• Specify how many days the study lasted.

• Indicate when each question was answered. For instance, was the subjective happiness scale completed only in the first session, only in the last session, during a random session, or multiple times?

• Specify the meaning of the 0-100 scales used. For instance, when asking “Since the last survey, I have been expressing love,” what were the labels on the endpoints? What were participants told in terms of what a score of 0 or 100 should be taken to mean?

• In the first sentence, “has been elevated to the status of a ‘Supreme emotion’” should say “has been elevated by some researchers…” It's not clear what it means to refer to one emotion as "Supreme" relative to others, and this position is not universal in the field.

• Reword this sentence to be grammatically correct (change vs. changes): “Higher inertia means that the level of loving feelings change more slowly, while lower inertia allows for quick regulation back to baseline.” Additionally, change the word “allows” to “refers to”.

• Add a “the” in the following sentence: “Conceptually, this means that more intensely participants expressed love, the stronger they were feeling loved, with this dynamic peaking around three hours.”

Reviewer #3: This is an interesting and important paper that illuminates new patterns for and between expressing love to another and feeling loved by another. The analyses appear to be well justified and described and the interpretations of them are accessibly communicated.

One area of murkiness that I would strongly advice be clarified is the term "felt love" which is ambiguous as to whether it reflect the love one person feel FROM another or FOR another. I understand that the author intend this variable to reflect the former (FROM another) yet in affective science, it is routine to describe a single emotional state in terms of the degrees to which it is FELT and/or EXPRESSED. Readers steeped in that tradition are likely to interpret "felt love" as love felt (but not necessarily expressed) FOR another.

**Do you want your identity to be public for this peer review?** For information about this choice, including consent withdrawal, please see our Privacy Policy

Reviewer #1: No

Reviewer #2: No

Reviewer #3: No

---

## [Author Response · Author response to Decision Letter 1]

31 Mar 2025

Please see Response to Reviewers file attached to this submission.

---

## [Editor Report · Decision Letter 1]

How much we express love predicts how much we feel loved in daily life

PONE-D-24-58443R1

Dear Dr. Oravecz,

We’re pleased to inform you that your manuscript has been judged scientifically suitable for publication and will be formally accepted for publication once it meets all outstanding technical requirements.

Thank you for amending all changes, notes by the Reviewers. You did an excellent job and I'm happy to accept your paper.

Kind regards,

Marta Kowal

Academic Editor

PLOS ONE

---

## [Editor Report · Acceptance letter]

PONE-D-24-58443R1

PLOS ONE

Dear Dr. Oravecz,

I'm pleased to inform you that your manuscript has been deemed suitable for publication in PLOS ONE. Congratulations! Your manuscript is now being handed over to our production team.

Kind regards,

on behalf of

Dr. Marta Kowal

Academic Editor

PLOS ONE